# Domain adaptation model for retinopathy detection from cross-domain OCT images

**Jing Wang**[1,2]                                                         CRYSKING_WJ@163.COM
[1] *University of Science and technology of China, Hefei 230026, China*
[2] *Jiangsu Key Laboratory of Medical Optics, Suzhou Institute of Biomedical Engineering and Technology, Chinese Academy of Sciences, Suzhou 215263, China*

**Yiwei Chen**[2]                                                         YIWEI.CHEN@SIBET.AC.CN
**Wanyue Li**[1,2]                                                        WANYUELI93@126.COM
**Wen Kong**[1,2]                                                         KONGWEN_WORK@163.COM
**Yi He**[2]                                                              HEYI@SIBET.AC.CN
**Chuihui Jiang**[*3]                                                     CHHJIANG70@163.COM
[3] *Department of ophthalmology and Vision Science, Eye and ENT Hospital, Fudan University, Shanghai 200031, People's Republic of China*
**Guohua Shi**[*2,4]                                                      GHSHI_LAB@126.COM
[4] *Center for Excellence in Brain Science and Intelligence Technology, CAS*

## Abstract

A deep neural network (DNN) can assist in retinopathy screening by automatically classifying patients into normal and abnormal categories according to optical coherence tomography (OCT) images. Typically, OCT images captured from different devices show heterogeneous appearances because of different scan settings; thus, the DNN model trained from one domain may fail if applied directly to a new domain. As data labels are difficult to acquire, we proposed a generative adversarial network-based domain adaptation model to address the cross-domain OCT images classification task, which can extract invariant and discriminative characteristics shared by different domains without incurring additional labeling cost. A feature generator, a Wasserstein distance estimator, a domain discriminator, and a classifier were included in the model to enforce the extraction of domain invariant representations. We applied the model to OCT images as well as public digit images. Results show that the model can significantly improve the classification accuracy of cross-domain images.

**Keywords:** domain adaptation, adversarial learning, OCT images, retinopathy detection.

## 1. Introduction

The macular is a vital organ of the human body and its degeneration can result in visual impairment and also blindness. Timely screening and early treatment can effectively reduce the blindness rate (van Velthoven et al., 2007). Optical coherence tomography (OCT), which can image retinal structures in vivo, has been widely applied in diagnostic ophthalmology owing to its ease of use, lack of ionizing radiation, and high resolution (Lang et al., 2013); there were more than 5 million OCT acquisitions in the US in 2014 (Wang et al.,

---

* Corresponding author

2016). However, the large amount of data creates a burden for doctors to manually evaluate individual images. Recent developments in computer-aided diagnostic systems (CADSs) have aided in retinopathy diagnosis and reduced the workload for clinicians.

Nonetheless, there are several OCT devices developed by different manufactures that vary in many aspects including imaging mode, image processing algorithm, hardware components, etc. Consequently, images captured from different devices have different signal distribution. Figure 1 shows two OCT images captured from different devices and their grayscale histogram, where the first is from Cirrus (Carl Zeiss Meditec, Inc., Dublin, CA) and the second is from Spectralis (Heidelberg Engineering, Heidelberg, Germany). Analyzing the images reveals that their appearance and signal distribution are quite different. It might be easy for experienced clinicians to examine the images with different signal distribution, but CADSs' performance declines when the test data are under a different distribution compared to the training data. It is an alternative to recollect and manually label new domain data and retrain another model for this domain, but it is time-consuming and expensive, particularly for medical images, where data are limited and collected from different devices. Therefore, methods that can learn from the source dataset and adapt to new target domain data, without additional labeling, are highly desirable.

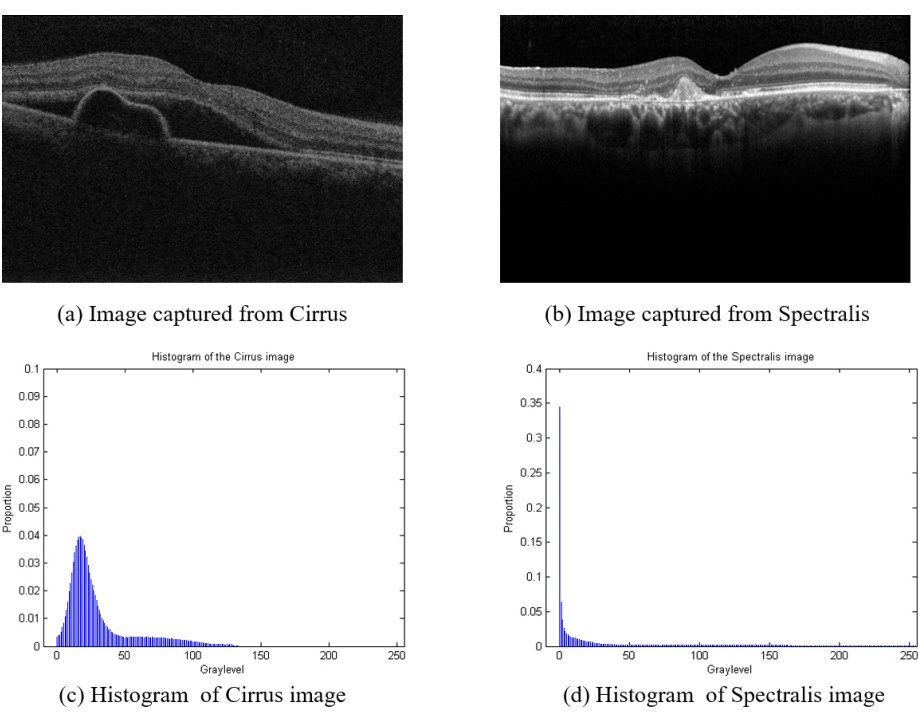

Figure 1: Examples of images and image histograms from different devices: (a) (c) Cirrus; (b) (d) Spectralis

In this study, we proposed a generative adversarial network (GAN)-based approach to reduce domain discrepancy in different modalities of OCT images by learning the domain invariant representations.

GANs have attracted lots of attention in recent years because they can generate target images from random signals by training networks in an adversarial manner. the most common GAN comprises a generator and discriminator; the generator aims to generate real-image-like images to deceive the discriminator, which is trained to distinguish the generated image from the real image. The domain adaptation can be considered as generative adversarial learning, as its generator learns to minimize the domain discrepancy distance while its discriminator tries to distinguish the domain of the original input. However, when the discriminator can perfectly distinguish target from source representations, the gradient vanishing problem will happen. Inspired by (Shen et al., 2017), we applied Wasserstein distance to provide more stable gradients in the adversarial learning process.

Several adversarial learning methods have been proposed to solve the domain shift problem in medical images (Chen et al., 2019b; Kamnitsas et al., 2017; Zhang et al., 2019; Ren et al., 2018; Romo-Bucheli et al., 2020), where (Romo-Bucheli et al., 2020) proposed a cycleGAN model to reduce covariate shift in OCT imaging for fluid segmentation task by synthesizing translated images. However, according to our knowledge, there is no research about domain adaptation on retinopathy detection from OCT images. The OCT devices that are commonly used in clinics come from different manufactures, such as Cirrus, Optovue, Topcon and Spectralis, thus domain shifts exited widely in the images, which heavily hindered the popularization of CADSs for retinopathy detection. In this work, we propose a domain adaptation model for OCT images, namely DAOCT. This model consists of a generator, two discriminators and a classifier, where discriminators estimate the discrepancy between the source and target features, the generator is optimized to minimize the estimated discrepancy in an adversarial manner, and the classifier is trained to detect retinopathy. A Wasserstein-distance based adversarial loss is designed to effectvely train this model. Finally, the generator can extract domain invariant features and the classifier can recognize abnormal OCT images from any domain. We have tested this model on our custom retina OCT images dataset and the public MNIST-USPS dataset pair. The results demonstrate that the DAOCT significantly improves the classification accuracy of retinopathy detection and digits recognition compared with other existing representation learning-based approaches.

## 2. Related works

Domain adaptation is an effective way to solve the shortage of label information, which is expensive and time-consuming to gather, especially for medical images. Model trained on the labeled data from one device can work on a new dataset with different distributions by reducing the domain dependencies between two dataset. There were lots of great domain adaptation works have been done recent years. A deep domain confusion (DDC) method was proposed (Tzeng et al., 2014) to minimize the divergence between two distributions by minimizing the maximum mean discrepancy (MMD) metric (Gretton et al., 2012). The MMD is a nonparametric metric that measures the distribution divergence between the mean embedding of two distributions in reproducing kernel Hilbert space (RKHS). Another excellent work was the DeepCORAL (Sun et al., 2016), which extends the correlation alignment (CORAL) method (Sun et al., 2016) to DNN to learn a nonlinear transformation that aligns correlations of layer activations.

Some other studies apply adversarial objectives to remove the domain discrepancy. (Shen et al., 2017) proposed Wasserstein distance guided representation learning (WDGRL) to reduces the distance between representations by minimizing the empirical Wasserstein distance in an adversarial manner. (Tzeng et al., 2017) proposed adversarial discriminative domain adaptation (ADDA) to reduce the domain distance by applying a standard GAN loss. Multilinear conditioning and entropy conditioning were applied to build the conditional domain adversarial networks (CDANs) by (Long et al., 2018) to better align different domains. Joint domain alignment and discriminative feature learning was proposed by (Chen et al., 2019a) to ensure that the domain invariant features obtain better intra-class compactness and inter-class separability which can significantly mitigate the domain shift.

Owing to the excellent performance of domain adaptations on the unsupervised domain shift mitigation, some domain adaption architectures have been applied on medical images where domain discrepancy is present and label information is difficult to acquire. A domain adaptation paradigm was proposed by applying an adversarial objective to adapt the Gleason score prediction model learned from annotated prostate whole-slide images (WSIs) to other unlabeled prostate WSIs (Ren et al., 2018). The noise adaptation GAN (NAGAN) was proposed by (Zhang et al., 2019) to solve the domain adaptation issue in OCT and ultrasound images by training a NAGAN to transfer the noise style from source images to that in the target domain; the NAGAN was trained in an adversarial manner with two discriminators. A synergistic image and feature adaptation (SIFA) architecture was proposed by (Chen et al., 2019b) to achieve cross-modality medical image segmentation of cardiac structures. (Kamnitsas et al., 2017) achieved brain lesion segmentation from MR images acquired using different scanners and imaging protocols.

In this study, we build a model to extract domain invariant representations from retina OCT images captured from different settings. Inspired by the WDGRL, we combined the Wasserstein distance with the domain distance of source and target representations to stably reduce the domain discrepancy. The evaluation results demonstrate the efficiency of our proposed method.

## 3. Method

The proposed DAOCT aims to extract invariant features to the covariate shift between OCT images captured from different domains. In this section, methods used to train this model are introduced.

### 3.1. Overview

Initially, we define a set of OCT images, annotated as normal or abnormal by experienced clinicians, captured from Cirrus as the source domain. The images in source domain distribution are labeled as $X^s = \{(x_i^s, y_i^s)\}_{i=1}^{N^s}$, where $y_i^s$ is one-hot vector denoting whether the retina is healthy or not and $N_s$ is the number of images. A set of OCT images captured from Heidelberg were defined as target domain and followed the distribution $T$. The target domain contains $N_t$ unlabeled OCT images and is defined as $X^t = \{x_j^t\}_{j=1}^{N^t}$. Unless otherwise specified, we define the symbols $s$ and $t$ as the source and target domain, respectively, in this manuscript and $s \in R_{x^s}$ and $t \in R_{x^t}$, where $R$ represent marginal data distribution.

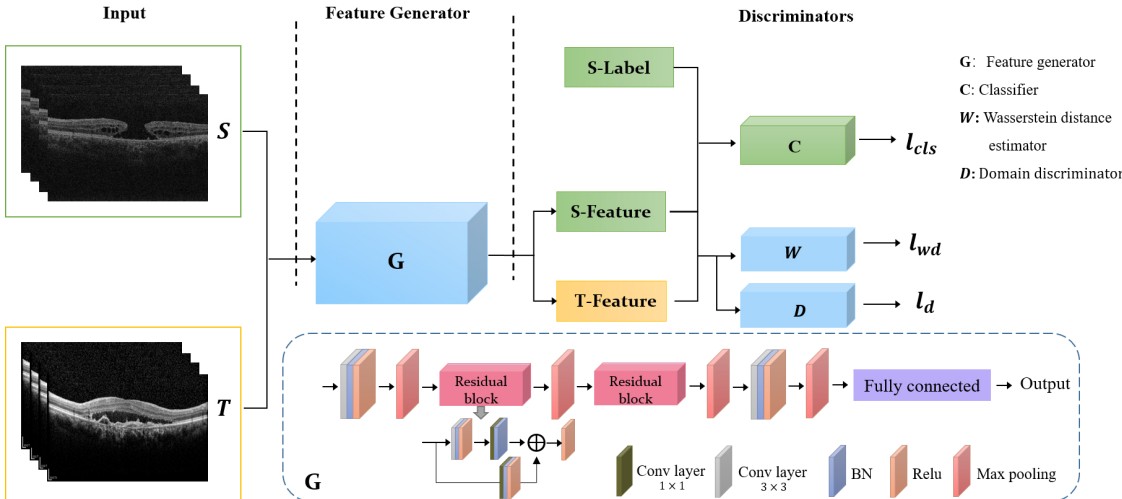

Figure 2: Architecture of DAOCT

The proposed DAOCT consists of a feature generator, a classifier, a Wasserstein distance estimator and an adversarial discriminator. The generator aims at domain invariant feature extraction, the Wasserstein distance estimator and the domain discriminator aim at minimizing the representation discrepancies and the classifier aims at screening retinopathy in images from both $s$ and $t$. The DAOCT architecture is presented in Figure 2. The feature generator, denoted as $G$, attempts to map images from each domain into a latent representation $z = G(x)$, which is expected to be domain invariant and category informative. The extracted representations are fed to the Wasserstein distance estimator and the domain discriminator, denoted as $W$ and $D$, respectively. The $W$ is used to evaluate the Wasserstein distance between source and target representations as the Wasserstein distance can help avoid the gradient vanishing problem and enable a more stable training process. $D$ is used to distinguish the original domain from its input. The classifier, denoted as $C$, attempts to classify the images from the target domain into normal and abnormal categories. The detailed information of the aforementioned architecture is as follows:

### 3.1.1. Feature generator

The feature generator $G$ is designed for mapping an input image from either the source or the target domain to a latent representation as follows:

$$z = G(x), x \in X^s \cup X^t, \tag{1}$$

where $G$ is a CNN with parameters denoted as $\theta_g$. For simplification, the latent representation of any source or target domain is denoted as $z^s = G(x^s)$ and $z^t = G(x^t)$, respectively.

### 3.1.2. Discriminator

After the extraction of latent representations from the feature generator, we use the $W$ and $D$ to estimate the distance between the source and target domain representation, specifically,

$W$ is applied to evaluate the Wasserstein distance and $D$ is applied to distinguish which domain the representation belongs to.

The Wasserstein distance was first applied to a GAN by (Arjovsky et al., 2017), helping to stablilize the GAN training. In the present study, we used the Wasserstein distance to estimate the distance between representation distributions $R_{z^s}$ and $R_{z^t}$, which can be computed by Equation (2).

$$W_1(R_{z^s}, R_{z^t}) = \sup_{\|f_w\|_L \leq 1} \mathbb{E}_{R_{z^s}}[W(z)] - \mathbb{E}_{R_{z^t}}[W(z)] \tag{2}$$

According to (Shen et al., 2017), if the parameterized family of $W$ are all 1-Lipschitz, then the empirical Wasserstein distance can be approximated by maximizing the domain critic loss $L_{wd}$ with respect to the parameter $\theta_w$, as follows:

$$L_{wd}(x^s, x^t) = \frac{1}{N^s} \sum_{x^s \in X^s} W(G(x^s)) - \frac{1}{N^t} \sum_{x^t \in X^t} W(G(x^t)) \tag{3}$$

Regarding the enforcing of the Lipschitz constraint, as (Gulrajani et al., 2017) highlighted, the weight clip proposed in (Arjovsky et al., 2017) causes capacity underuse and gradient vanishing or exploding problems. Thus, they proposed a more reasonable alternative by enforcing gradient penalty $L_{grad}$ for the domain critic parameter $\theta_w$, as follows:

$$L_{grad}(\hat{z}) = (\|\nabla_{\hat{z}} W(\hat{z}))\|_2 - 1)^2 \tag{4}$$

The gradients are penalized at feature representations $\hat{z}$ which are defined not only at the source and target representations pair as well as the random points along the straight line between the pair. The Wasserstein distance between two representations can be estimated by solving the problem:

$$\max_{\theta_w} \{L_{wd} - \alpha L_{grad}\} \tag{5}$$

Domain discriminator $D$ is trained to classify which domain is the origin of the data point, and can be optimized by a standard classification loss $L_{A^D}(x^s, x^t)$ with respect to the parameter $\theta_d$, where the labels indicate the origin domain, defined as:

$$L_{A^D}(x^s, x^t) = -\mathbb{E}_{x^s \sim X^s}[\log D(G(x^s))] - \mathbb{E}_{x^t \sim X^t}[1 - \log D(G(x^t))] \tag{6}$$

Finally, we can estimate the discrepancy between two domains by solving the problem

$$\max_{\theta_w, \theta_d} \{L_{wd} - \alpha L_{grad} + L_{A^D}\} \tag{7}$$

where $\alpha$ is a balancing coefficient. As the generator aims to extract domain invariant features to fool $D$, it can be optimized by minimizing the $L_{A^M}$ with respect to the parameter $\theta_g$:

$$L_{A^M} = -L_{A^D} \tag{8}$$

The Wasserstein distance is applied to estimate the domain discrepancy, and it is continuous and differentiable in almost any situation. Thus, we can first train $W$ and $D$ to

optimality (Shen et al., 2017). Fixing the optimal parameters of $D$ and $W$ and minimizing the distance between representations, the feature generator can learn the feature representations with reduced domain discrepancy. This representation learning process can be achieved by solving the adversarial problem

$$\min_{\theta_g}\{L_{A^M} + \lambda \max_{\theta_w,\theta_d}[L_{wd} - \alpha L_{grad} + L_{A^D}]\} \tag{9}$$

where $\lambda$ is a balancing coefficient and $\alpha$ should be set as 0 when performing the minimization because the gradient penalty should not affect the feature generator training process. Finally, the feature generator can extract domain invariant representations.

### 3.1.3. Classifier

As the feature generator can only extract domain invariant representations, it can not achieve our final goal of training a high-performance classifier for the target domain. Hence, several additional layers were added to serve as a classifier ($C$). As the label of the target domain is unavailable, $C$ is trained with the source representations that have elaborated label information. The classifier loss is calculated using the standard supervised loss which is defined as follows:

$$L_{cls}(x^s, y^s) = -\frac{1}{N^s}\sum_{i=1}^{N^s}\sum_{k=1}^{l}\mathbb{I}(y_i^s = k) \cdot \log C(G(x_i^s))_k, \tag{10}$$

where $l$ is the number of classes.

### 3.2. Overall objective

Finally, combining the aforementioned loss, we can obtain our overall objective function:

$$\min_{\theta_g,\theta_c}\{L_{cls} + L_{A^M} + \lambda \max_{\theta_w,\theta_d}[L_{wd} - \alpha L_{grad} + L_{A^D}]\} \tag{11}$$

where $\lambda$ is a balancing coefficient and $\alpha$ should be set as 0 when optimizing the minimum operation.

The detailed algorithm of the training process is given in Algorithm 1.

## 4. Experiment

The efficiency of the proposed method is evaluated on a public dataset and a private dataset. The evaluation results demonstrate that the proposed method shows better performance than some of the state-of-the-art methods, which also aimed at solving the domain shift problem by extracting the domain invariant representations, including WDGRL (Shen et al., 2017),DANN (Ganin et al., 2015), CADN (Long et al., 2018), JDDA-CORAL (Chen et al., 2019a), JDDA-MMD (Chen et al., 2019a).

---

**Algorithm 1:** Domain adaptation for optical coherence tomography

---

**Input:** The source domain sample $x^s$ and source category label $y^s$, target domain
sample $x^t$ without label.

**Data:** minibatch size $m$; critic and discriminator training step: $n$; classifier and
generator training step: $N$; coefficient: $\alpha$; critic and discriminator learning rate:
$\gamma_1$; classifier and feature generator learning rate: $\gamma_2$.

Initialize feature generator, domain critic, discriminator, classifier with random weights
$\theta_g, \theta_w, \theta_a, \theta_c$ ;

**for** $T \leftarrow 1$ **to** $N$ **do**

    Acquire source minibatch $\{x_i^s, y_i^s\}_{i=1}^m$, target minibatch $\{x_j^t\}_{j=1}^m$ from $X^s$ and $X^t$ ;

    **for** $t \leftarrow 1$ **to** $n$ **do**

        $Z^s = G(x^s), z^t = G(x^t)$;

        Sample random points z along straight lines between $z^s$ and $z^t$ pairs;

        $\hat{z} \leftarrow \{z^s, z^t, z\}$;

        $\theta_w \leftarrow \theta_w + \gamma_1 \nabla_{\theta_w}[L_{wd}(x^s, x^t) - \alpha L_{grad(\hat{z})}]$;

        $\theta_d \leftarrow \theta_d + \gamma_1 \nabla_{\theta_d}[L_{A^D}(x^s, x^t)]$;

    **end**

    $\theta_c \leftarrow \theta_c - \gamma_2 \nabla_{\theta_c} L_{cls}(x^s, y^s)$;

    $\theta_g \leftarrow \theta_g - \gamma_2 \nabla_{\theta_g}[L_{cls}(x^s, y^s) + L_{A^M}(x^s, y^s) + L_{wd}x^s, y^s]$;

**end**

Return $G, C$;

---

### 4.1. Dataset

We tested our method on two datasets:

**Digits recognition datasets.** They are the most widely used benchmark datasets and contain digit images ranging from 1 to 10 with different styles. We applied the MNIST-USPS pair to evaluate our method. In this experiment, we assigned the MNIST (M) and USPS (U) as the source and target domain, respectively, and adopted the standard training set (60,000 pictures for M, 7,291 pictures for U), and the valuation dataset (10,000 pictures for M, 2,007 pictures for U). The sizes of M and U were unified to 28×28 pixels, and transferred to gray images.

**Retinal OCT images.** This custom dataset consists of retinal images captured from two different OCT devices, Cirrus (Carl Zeiss Meditec, Inc., Dublin, CA) and Spectralis (Heidelberg Engineering, Heidelberg, Germany), both of which are widely applied in clinics. Images captured from these devices are different in many aspects, such as signal distributions, noise style, shown in Figure 1, resolutions (1536×1024 pixels for Cirrus images, 765 × 496 pixels for Spectralis images) and so on. We applied the images captured from Ciruss (denoted as Z) as the source domain because they have detailed label information which was acquired in (Wang et al., 2019). The Z consisted of 7,096 abnormal and 5,738 normal samples captured from 710 subjects and labeled by two specialists, where 472 images were randomly chosen as the evaluation set and the remaining were assigned as training set (scans from the same person were kept in the same set). Another set of retinal OCT images captured from the Spectralis (denoted as H) was assigned as target domain. The H

consisted of 12,427 images captured from 678 subjects, same as the source domain dataset. 1,007 images of H were randomly selected as the evaluation set, which were labeled by a specialist with more than 10 years' experience in clinical retinopathy detection experience, and the remaining images, with no label information, were used to train the model. To prevent overfitting, the training dataset of the source domain, which was used to train the retinopathy classifier, was augmented by horizontal mirroring and contrast enhancement. The contrast enhancement is defined by Equation (12), where $I$ denotes the destination image, $i$ is the source image, and $v$ ($v = 10$ in this study) indicates the degree of contrast enhancement. The image sizes of Z and H were unified to $112 \times 112$ pixels and the images were converted to grayscale to reduce the computation complexity.

$$I = \log 2(1 + v * i)/\log 2(v + 1) \tag{12}$$

## 4.2. Network architecture

The overall network architecture is provided in Figure 2, consisting of $G$, $C$, $W$ and $D$. $G$, the core architecture of our network, is a convolutional neural network as show in Figure 2 which applies residual block to ease the gradients flow. Specifically, only the first and the last max-pooling layer were available for the digits classification task as the input size was different for the two tasks. The $C$ consists of a fully connected (FC) layer with two outputs, denoting whether the retina is sick or not. Both $W$ and $D$ consist of two FC layers, where $W$ has one output to estimate the Wasserstein distance between two domains and $D$ has two outputs to denote the domain of the original input.

## 4.3. Experiment setup

We compared our method with some related works, such asWDGRL (Shen et al., 2017), DANN (Ganin et al., 2015), CADN (Long et al., 2018), JDDA-CORAL (Chen et al., 2019a), JDDA-MMD (Chen et al., 2019a), which also focused on learning the domain invariant representations to reduce the domain discrepancy by symmetric ways. The JDDA proposed a two-stream network and combined a center loss in the domain adaptation process to make the intra-class representations more compact and increase the distance between inter-class representations. In their work, the authors have proved that the combination of JDDA with CORAL and MMD enables the achievement of a better performance. Thus we only compared our method with the JDDA-CORAL and JDDA-MMD. There are other effective domain adaptation methods, such as Pixel-GAN (Bousmalis et al., 2017), DupGAN (Hu et al., 2018) and the NAGAN (Zhang et al., 2019), which are not used in the comparison because they focus on synthesizing target-like images according to source images to reduce the domain shift; our approach can be easily integrated into these works.

Our experiments were implemented via TensorFlow and trained with Adam optimizer. Following (Long et al., 2013), it is impossible to perform cross validation in this experiment, since labeled and unlabeled data are sampled from different distributions, we found the best results of all methods through grid search on the hyper-parameter space. The batch size was set as 64, 32 for each domain, and the learning rate was fixed as $10^{-4}$. In order to equally compare all the methods, the classifier was combined with all comparison methods

to perform the classification task, the feature generators of all comparison methods were replaced by the proposed feature generator.

The proposed approach could be implemented according to Algorithm 1. Additional training steps $n$ were selected for additional optimization for the $W$ and $D$ to achieve the Nash Equilibrium faster. In our experiment, $n$ was set as 10. Following (Shen et al., 2017), we penalized the gradients at the representations of two domains and random points along the straight line between the source and target pairs; the coefficient $\alpha$ was set as 10.

## 4.4. Results and discussion

We evaluated the proposed architecture on the digits dataset and the retina OCT images dataset, as shown in Table 1. To better demonstrate the efficiency of the trained models, we tested them on both target and source evaluation set (the result of the source evaluation set was reported in parentheses in Table 1). The source-only model means that the model was only trained on labeled source dataset. It is evident that our method outperforms all methods compared on both classification tasks. The WDGRL, with more reliable gradient, obtained better results than the JDDA-based methods. It can be inferred that the combination of Wasserstein distance and adversarial loss can better enforce the extraction of the domain invariant features.

It should be noted that when testing the source evaluation set, our proposed method only has a minor accuracy reduction while other domain adaptation methods have a larger accuracy reduction compared with the source only method, especially on the retina OCT images. It demonstrates that when performing domain adaptation, the WDGRL and JDDA-based methods tend to extract target domain representations, and our proposed method tries to extract domain invariant representations. Thus we can conclude that our proposal is more effective in removing the domain shift of OCT and digit images. Furthermore, we found that the target images classification accuracy of source-only model on the MNIST $\rightarrow$ USPS task significantly improved compared with that reported in existing works, i.e. UDAR (Hou et al., 2019), Pixel-GAN (Bousmalis et al., 2017), DupGAN (Hu et al., 2018). These methods are not included into the former comparison as they focused on synthesizing target-like images according to source images to reduce the domain shift rather than learning the domain invariant representations. However, these works have also carefully designed the architecture of the feature generator to extract effective representations. The result is shown in Table 2. It can be noticed that our method obtains an improvement on the accuracy, i.e. nearly 32% higher than the UDAR (Hou et al., 2019) that also applied residual architecture in their model. This result confirmed that our feature generator is more effective in extracting category informative features, which can overcome part of the domain shift before domain adaptation training.

**Ablation Study** We evaluate the contributions of the discriminator ($L_{A^D}$) and the Wasserstein distance estimator ($L_{wd}$) separately. The result was shown in Table 3. It can be found that both the $L_{wd}$ and $L_{A^D}$ work effectively in reducing the domain distance, where $L_{wd}$ works a little better when they were applied alone. But according to the table, the result can be further improved when applying these two components at the same time, it confirmed the effectiviness of the proposed loss strategy. We also tested the effectiviness of the proposed the feature generator architecture by comparing it with the multi-layer

Table 1: Evaluation results (accuracy %) of several domain adaptation models on target datasets. (The evaluation results on the source dataset is reported in parentheses)

| Method | MNIST→USPS | Cirrus→Spectralis |
|---|---|---|
| Source only | 0.9612(0.9939) | 0.8669(0.947) |
| WDGRL | 0.9756(0.9908) | 0.9374(0.872) |
| JDDA_CORAL | 0.9314(0.9798) | 0.9156(0.8671) |
| JDDA_MMD | 0.9368(0.985) | 0.9255(0.8575) |
| CADN | 0.9696(0.9958) | 0.8292(0.7223) |
| DANN | 0.9273(0.9953) | 0.8699(0.6631) |
| DAOCT | **0.9804**(0.9914) | **0.9553**(0.9307) |

Table 2: Evaluation results of several source-only models on target dataset

| Method | MNIST→USPS |
|---|---|
| UDAR (Hou et al., 2019) | 0.634 |
| Pixel-GAN (Bousmalis et al., 2017) | 0.789 |
| DupGAN (Hu et al., 2018) | 0.8675 |
| DAOCT (proposed) | 0.9612 |

Table 3: Effectives of each key component in DAOCT, evaluation accuracy (%) on target dataset. 'FG' means feature gennerator proposed in this study, and multi-layer perceptron is set as default feature generator

| Method | Source only | $L_{wd}$ | $L_{A^D}$ | FG | Accuracy |
|---|---|---|---|---|---|
| MNIST→USPS | ✓ | | | | 0.9301 |
| | | ✓ | | | 0.9656 |
| | | | ✓ | | 0.9371 |
| | | ✓ | ✓ | | 0.9667 |
| | | ✓ | ✓ | ✓ | **0.9804** |
| Cirrus→Spectralis | ✓ | | | ✓ | 0.8669 |
| | | ✓ | | ✓ | 0.9374 |
| | | | ✓ | ✓ | 0.9359 |
| | | ✓ | ✓ | | 0.8758 |
| | | ✓ | ✓ | ✓ | **0.9553** |

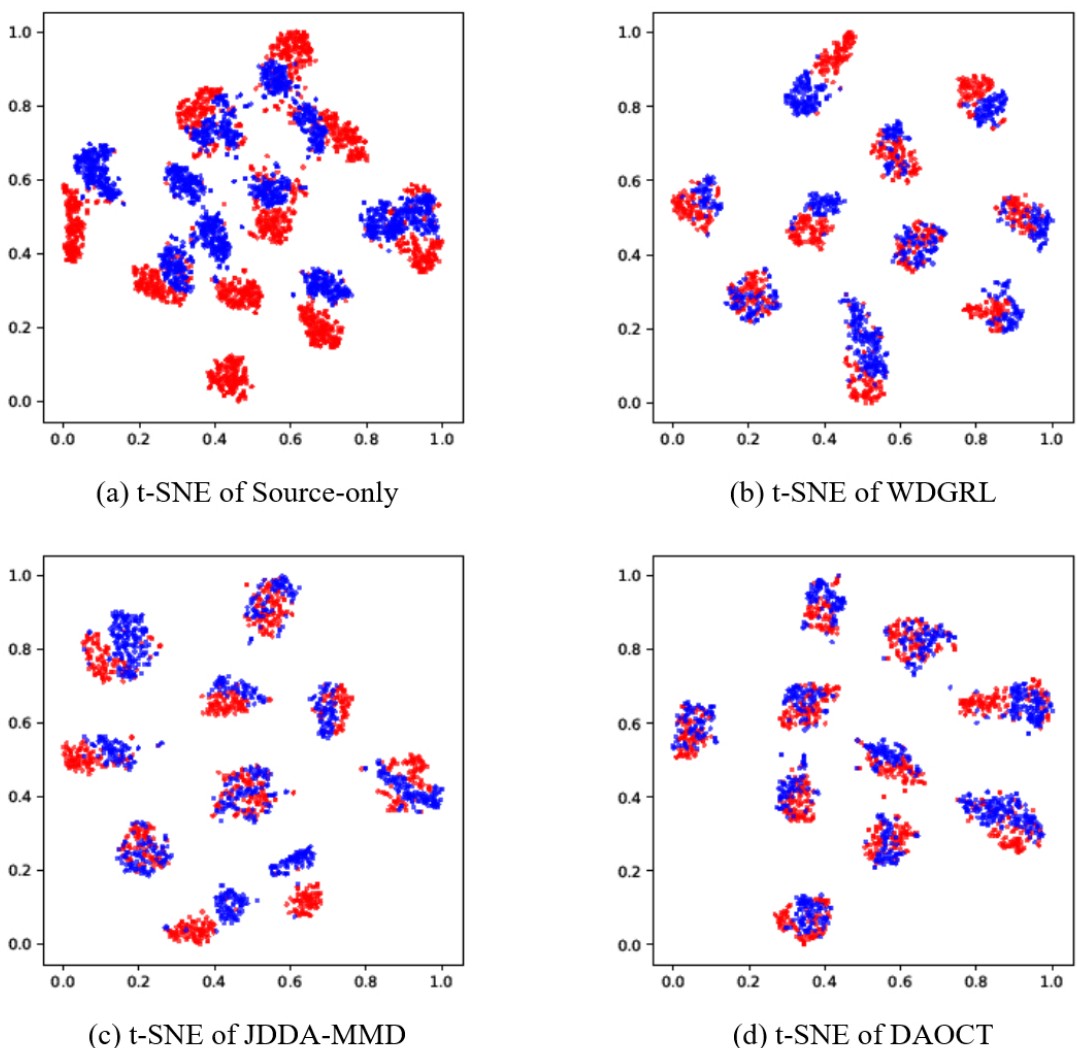

(a) t-SNE of Source-only

(b) t-SNE of WDGRL

(c) t-SNE of JDDA-MMD

(d) t-SNE of DAOCT

Figure 3: t-SNE of digits classification

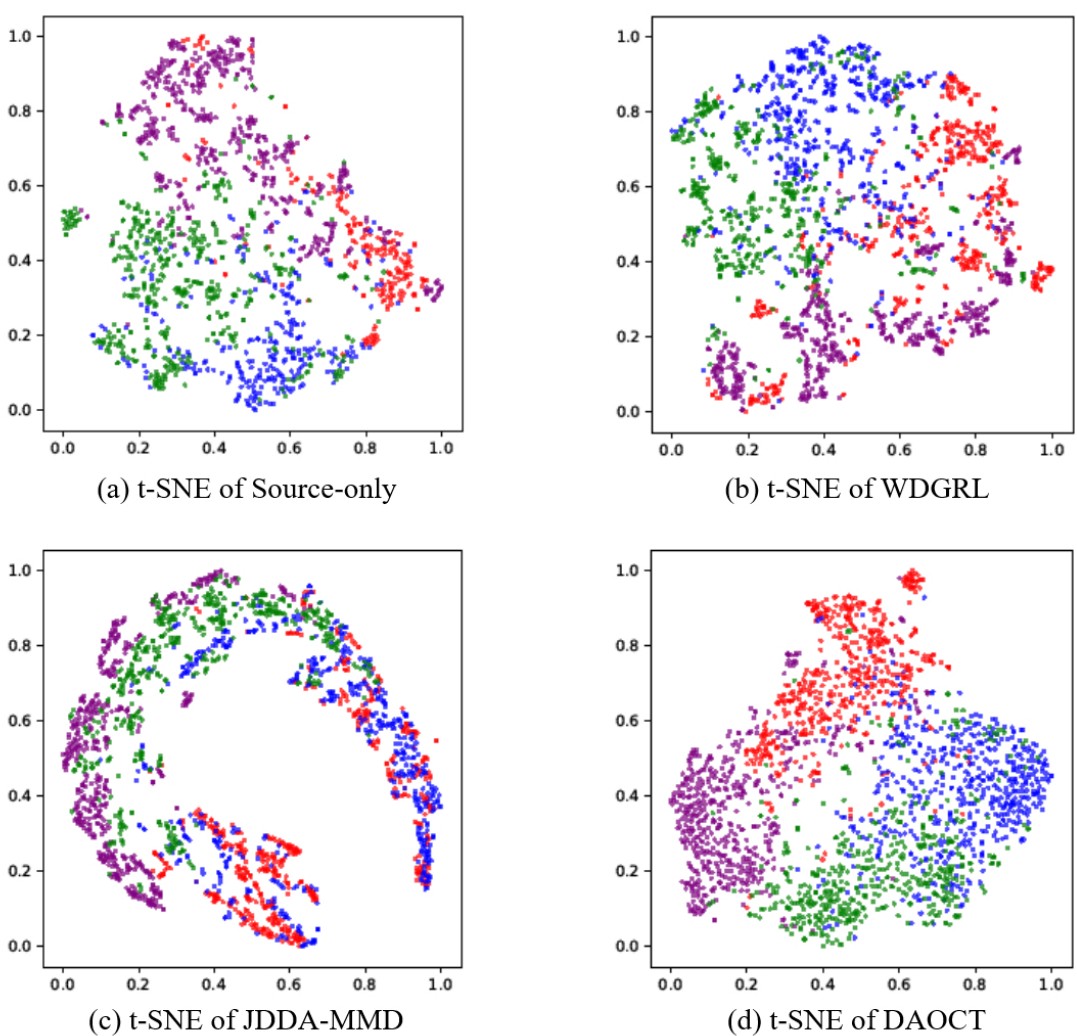

(a) t-SNE of Source-only    (b) t-SNE of WDGRL

(c) t-SNE of JDDA-MMD    (d) t-SNE of DAOCT

Figure 4: t-SNE of OCT images classification

perceptron. We applied the proposed feature generator and multi-layer perceptron on two tasks under the same loss strategy as shown in Table 3. It can be inferenced from the result that the proposed feature generator did a beeter job in extracting domain invariant and discriminative features as the accuracies were significantly improved when applying the custom feature generator on both two tasks.

**Feature visualization** We plot the t-SNE visualization of the digits classification tasks to analyze the representations distributions, as shown in Figure 3, where the red and blue spots represent the source and target domain in separate, respectively, and each cluster represents a category. According to the figures, we can observe that the WDGRL and JDDA is effective in reducing the domain shift in some extent, but the performances are not so perfect as the source and target domains of some clusters are still far apart. We can observe that the blue and red spots of all the clusters align better in Figure 3(d), which means the proposed method was more effective in reducing the domain distance of the MNIST and USPS datasets.

We also plot the t-SNE visualization to demonstrate the efficiency of the proposed model on the retina OCT images, as show in Figure 4, where the red and blue spots represent normal and abnormal samples of the source domain, resprectively, and purple and green spots represent the normal and abnormal samples of the target domain,respectively. It can be concluded that our proposed method obtains the best performance in reducing the representations' distance between the source and target domains as the distance between red (blue) and purple (green) spots is closer in Figure 4(d), which represents the t-SNE visualization of our method. The results also demonstrate that our proposed method can increase the distance between different categories in target domain as the distances between the green and purple spots are farther than the other comparison methods.

## 5. Conclusion

Domain shift is an important issue in machine learning, particularly for medical images which are often captured from different devices. In this paper, we proposed an adversarial learning-based network, which consists of a feature generator, two discriminators and a classifier, to extract domain invariant and category informative representations from OCT images having different signal distributions. The Wasserstein distance was combined with the adversarial loss to optimize the network effectively. We tested our network on the $Z \rightarrow H$ and $M \rightarrow U$ tasks and compared the results with those of some related works, demonstrating the effectiveness of the proposed network architecture and loss. The feature visualizations indicate that our method can approximate representations from different domains while maintaining the distance among categories. The test results on source evaluation datasets provide further evidence that our method can effectively reduce the domain shift.

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
