# OpenReview forum: "Domain adaptation model for retinopathy detection from cross-domain OCT images"
_MIDL.io/2020/Conference — MIDL 2020_

### Official Review · AnonReviewer2 · 2020-03-13
**A simple domain adaptation method with good results achieved on a new application**

**Rating:** 3
**Confidence:** 4
**Recommendation:** Poster

**Summary:**

This paper proposes a domain adaptation method for retinopathy detection from OCT images. It is based on a general domain adaptation framework aiming to learn domain-invariant features with a straightforward combination of a Wasserstein distance estimator and a domain discriminator. Evaluation is conducted with public digits datasets and private OCT datasets with good results obtained.

**Strengths:**

- The paper tackles the problem of domain adaptation on a new application, i.e., retinopathy detection from OCT images.
- The presented method is validated on the public digits datasets and private OCT datasets.
- Good results are achieved.

**Weaknesses:**

- My major concern lies in the motivation of combining a Wasserstein distance estimator with a domain discriminator. Why could their combination contribute to extract more domain-invariant features so as to improve domain adaptation performance? The authors are suggested to justify about it.
- On a related note, ablation study of the Wasserstein distance estimator and the domain discriminator should be conducted, to analyze the contribution of the two components on extracting domain-invariant features.
- In Table 2, it is interesting that the evaluation results of source-only model are quite different from different methods. What could be the reason for that? The proposed method obtains quite good results of source-only model. It would be great to investigate what component in the model contribute to that.
- The implementations of other methods in comparison are not clearly described. For the comparison with other methods on the digits datasets, are the results directly referenced from their papers? For the comparison on OCT images, how are the methods in comparison implemented? What network architectures do they use?
- I found that the formulations and description of the Wasserstein distance estimator is very similar to that in WDGRL. The authors are suggested to reformulate it to avoid similarities.
- In Table 2, the reference format needs to be made consistent with other references.

**Justification Of Rating:**

Domain adaptation on retinopathy detection from OCT images has not been studied before. The proposed method is effective in extracting domain-invariant features and achieves good domain adaptation results.

**Paper Type:**

both

**Special Issue:**

no

---

> ### Author Response · Authors · 2020-03-26
> **Reply to reviewer's comments**
>
> We appreciate your effort in reviewing our paper. We clarify the issues as follows:
> Weaknesses:
>  - My major concern lies in the motivation of combining a Wasserstein distance estimator with a domain discriminator. Why could their combination contribute to extract more domain-invariant features so as to improve domain adaptation performance? The authors are suggested to justify about it.
>
> Reply: As the Wasserstein distance estimator can reduce the discrepancy between two domains by minimizing the Wasserstein distance, which can also prevent the model training from over-learning. But we found that the ability of Wasserstein distance in reducing the domain distance is limited, so we added a domain discriminator to directly supervise the domain distance, and the result demonstrates this combination is effective. We’ll explain this more explicit as suggested in the final version.
>
> - On a related note, ablation study of the Wasserstein distance estimator and the domain discriminator should be conducted, to analyze the contribution of the two components on extracting domain-invariant features.
>
> Reply: Thank you for your valuable advise, we’ll add an ablation study in our final manuscript to discussion the contribution of the Wasserstein distance estimator and the domain discriminator.
>
> - In Table 2, it is interesting that the evaluation results of source-only model are quite different from different methods. What could be the reason for that? The proposed method obtains quite good results of source-only model. It would be great to investigate what component in the model contribute to that.
>
> Reply: Thank you for your valuable advise. In Table 2, we compared some evaluation results of several different models that focus on modifying the model architecture. The results demonstrated that our model,which was designed in a residual style, got the best performance in extracting decisive information to detect retinopathy on OCT images. Because of the limited space, we will add some experiments to explain this in  appendix of the final version.
>
> - The implementations of other methods in comparison are not clearly described. For the comparison with other methods on the digits datasets, are the results directly referenced from their papers? For the comparison on OCT images, how are the methods in comparison implemented? What network architectures do they use?
>
> Reply: We would like to kindly remind that we have mentioned this in the last line of paragraph 2 of 4.3 section: “In order to equally compare all the methods, the classifier was combined to achieve the classification task, the proposed feature generator was applied as a basic network, and the parameters were set according to the original works.”.  We re-implemented the experiment of all mentioned methods on the digits datasets and reported the results. For the comparison on OCT images, the feature extractors of all the methods are the one we proposed, the discriminators of compared methods are build according to their original works.  We realized that the description might be  confused, we will explain it more explicit in the final version.
>
> - I found that the formulations and description of the Wasserstein distance estimator is very similar to that in WDGRL. The authors are suggested to reformulate it to avoid similarities.
>
> Reply: Thank you for your kind remind, we’ll reformulate it in the final version.
>
> - In Table 2, the reference format needs to be made consistent with other references.
>
> Reply: Thank you for your kind remind, we’ll correct it in the final version

---

### Official Review · AnonReviewer1 · 2020-03-13
**Reasonable paper**

**Rating:** 3
**Confidence:** 4

**Summary:**

Presenting a deep neural network to classify retinopathy in OCT images, the authors propose a domain adaptation method that combines an adversarial domain discriminator with a Wasserstein distance minimization. Experiments on OCT images and on handwritten digits suggest that this approach works better than alternative methods.

**Strengths:**

* The experiments seem well-designed, with a separate evaluation set.
* The method is compared with multiple competing methods, showing an improved performance on both problems.
* The application to retinopathy detection seems fairly novel.
* The authors provide some t-SNE-based analysis of the network output.

**Weaknesses:**

* Although there is a comparison with alternative methods, I would have liked to see an ablation study in which the authors compared versions of their own method. This would allow us to evaluate the contributions of the discriminator and the Wasserstein distance separately.
* The novelty of the methods is not extremely clear. How is this different from Shen et al. 2017?
* The authors refer to their feature extractor as a "generator". I don't think this is a term that fits here: a generator usually refers to a model that outputs something like an input image. I would suggest to just call this an encoder.
* The paper makes a sloppy impression at times, as if it has been put together in a hurry (see the detailed comments for some examples). It would be good to do a careful proofreading of the text.

**Detailed Comments:**

Some issues in the writing:
The section titles for sections "3. method" and "3.1 overview" are not capitalised, as is "method" in Table 1.
3.1, first line: what is "elaborate label information"?
3.1 last line before figure: what is R in $s \in R_{x^s}$ and $t \in R_{x^t}$?
4 first line: "a public and a private datasets"

In tables 1 and 2, it would be useful to note the meaning of the parentheses in the caption of the table, not just in the text. Perhaps this would fit as a second line in the left column?

I was a bit surprised by this sentence in section 4.3:
"Following the evaluation method for domain adaptation (Long et al., 2013), we found the best results of all methods through grid search on the hyper-parameter space."
It is obviously fine to follow earlier papers, but I would hesitate to call this *the* evaluation method for domain adaptation. Grid search is a pretty common method.

**Justification Of Rating:**

I think this is a reasonable paper. The application is interesting, the experiments, the method is evaluated on two datasets, there is some analysis of the results, and the method is compared with alternative approaches.

**Paper Type:**

both

**Questions To Address In The Rebuttal:**

The evaluation is now done in one direction only. Would you expect similar results if you would swap the source and target domains?

The method is named DAOCT, but which aspects of it are specific to OCT images?

**Special Issue:**

no

---

> ### Author Response · Authors · 2020-03-26
> **Reply to reviewer's comments**
>
> We appreciate your effort in reviewing our paper. We clarify the issues as follows:
> Weaknesses:
> * Although there is a comparison with alternative methods, I would have liked to see an ablation study in which the authors compared versions of their own method. This would allow us to evaluate the contributions of the discriminator and the Wasserstein distance separately.
>
> Reply: Thank you for your constructive comment, we’ll add an ablation study to discuss the contribution of the discriminator and the Wasserstei distance estimator in the final version paper.
>
> * The novelty of the methods is not extremely clear. How is this different from Shen et al. 2017?
>
> Reply: The WDGRL (Shen et al, 2017) reduces the discrepancy between two domains only by minimizing the Wasserstein distance. We used a domain discriminator in combination with the WDGRL to directly monitor the distance between representations of two domains. The result demonstrate that our architecture performed better in reduce the domain discrepancy.
>
> * The authors refer to their feature extractor as a "generator". I don't think this is a term that fits here: a generator usually refers to a model that outputs something like an input image. I would suggest to just call this an encoder.
>
> Reply: We call the feature extractor as a generator because we applied a adversarial network architecture to build our model, so we  empirically call the feature extractor as a generator. If it’s confusing, we’ll correct it as suggested in our final paper.
>
> * The paper makes a sloppy impression at times, as if it has been put together in a hurry (see the detailed comments for some examples). It would be good to do a careful proofreading of the text.
>
> Reply: We apologize for these errors, and thanks for the reviewer’s kind remind, we’ll check our paper carefully and correct the errors. A careful proofreading will be performed in the future.
>
> Questions To Address In The Rebuttal:
> 1. The evaluation is now done in one direction only. Would you expect similar results if you would swap the source and target domains?
>
> Reply: We agree that swapping source and target domains and do the same training will make our result more convincing, but we’d like to kindly remind that we proposed this model because we can’t acquire the label information of the target domain.
>
> 2. The method is named DAOCT, but which aspects of it are specific to OCT images?
>
> Reply: We call this model DAOCT because it’s the OCT images that drive us to propose this model, and we only applied this model on OCT images and hand-writing digits dataset (as a baseline), we are not sure if this model can perform well on other dataset, which will be solved in our future work.
>
> Detailed Comments: Some issues in the writing:
> -The section titles for sections "3. method" and "3.1 overview" are not capitalised, as is "method" in Table 1.
> Reply: Thank you for your kind remind, we’ll correct it in the final version.
>
> -3.1, first line: what is "elaborate label information"?
> Reply:  We proposed this model to solve the domain shift in retinopathy detection, so the “elaborate label information” means that all source images were labeled as normal or abnormal. We’ll add this to the final version paper.
>
> -3.1 last line before figure: what is R in $s \in R_{x^s}$ and $t \in R_{x^t}$ ?
> Reply: R means marginal data distributions here, we’ll add this to the final version.
>
> -4 first line: "a public and a private datasets"
> -In tables 1 and 2, it would be useful to note the meaning of the parentheses in the caption of the table, not just in the text. Perhaps this would fit as a second line in the left column?
> Reply: Thank you for your kind remind, we'll correct these errors in the final version.
>
> -I was a bit surprised by this sentence in section 4.3:
> "Following the evaluation method for domain adaptation (Long et al., 2013), we found the best results of all methods through grid search on the hyper-parameter space."
> It is obviously fine to follow earlier papers, but I would hesitate to call this *the* evaluation method for domain adaptation. Grid search is a pretty common method.
> Reply: We apologize for our confusing description. We referred the Long et al., 2013 in our paper, because this paper explained why cross-validation can’t be implemented in domain adaptation tasks: “it is impossible to perform cross validation, since labeled and unlabeled data are sampled from different distributions. ”. We’ll revise the description in the final version paper.

---

### Official Review · AnonReviewer4 · 2020-03-13
**Interesting application of domain adaptation to retinopathy classification**

**Rating:** 3
**Confidence:** 5
**Recommendation:** Poster

**Summary:**

This paper proposes a methodology to address a cross-domain retinopathy classification task. For doing so, a feature generator, a Wasserstein distance estimator, a domain discriminator, and a classifier were included in the model to enforce the extraction of domain invariant representations. The problem of designing methods for reducing the performance drop in OCT images from a different vendor is relevant for automated OCT imaging analysis.

**Strengths:**

-  The problem of designing methods for reducing the performance drop in OCT images from a different vendor is relevant for automated OCT imaging analysis. The proposed method is new and it was compared with up to date baseline methods.

**Weaknesses:**

The contribution of the paper is not clear. Also the feature visualization showcased at the results section is difficult to evaluate.

The paper does not cite research on the OCT cross-domain adaptation for different OCT devices. For instance, in “Reducing image variability across OCT devices with unsupervised unpaired learning for improved segmentation of retina” (Romo et al., 2019), the cross-domain adaptation for segmentation tasks was achieved by using the cycleGAN algorithm.

**Detailed Comments:**

“However, when the discriminator can perfectly distinguish the target from source representations, there is a gradient vanishing problems.” I think this is an oversimplification of GAN training difficulties. Additionally, the authors could also include a noisy loss signal, mode collapse, and the synchronization strength between the generator and discriminator.

- Please add details about the contrast enhancement used as data augmentation during training.

- Figures 3. and 4. are very difficult to analyze correctly. Also, Tsne is a non-deterministic visualization technique that generates different visualizations and depends heavily on the perplexity hyperparameter. Given these limitations, I would say it is not advisable to use these visualizations to demonstrate that the method reduces the representations’ distance. I would recommend using feature maps from the generator (last conv layer before Fully connected layer) to showcase such reduction.

MINOR COMMENTS
- Section "3. method" and "3.1. overview" are in lowercase
- “…into a latent representationz = ”
- “zwhich are defined not only at the source”
- “to achieve the Nash Equilibrium faster (change the description).”



**Justification Of Rating:**

The topic covered by the paper is relevant to the OCT imaging community. However, there are considerable gaps in the presentation of the results, description of the methods and analysis of the results.

**Paper Type:**

methodological development

**Questions To Address In The Rebuttal:**

- What would be the advantage of using the proposed domain adaptation in contrast to generating an unsupervised unpaired cycleGAN model and then feeding the classifier with “translated” images?

- In Algorithm 1, what distribution follows the random weights in the initialization process?

- Table 1 caption does not describe which metric is showcased. What is the meaning of the values in parentheses? This information is in the text but not in the caption of the Table.

**Special Issue:**

no

---

> ### Author Response · Authors · 2020-03-26
> **Reply to reviewer's comments**
>
> Thank you for your time and efforts in reviewing our paper. We clarify the issues as follows:
> Questions To Address In The Rebuttal:
>  - What would be the advantage of using the proposed domain adaptation in contrast to generating an unsupervised unpaired cycleGAN model and then feeding the classifier with “translated” images?
>
> Reply: The cycleGan model focused on synthesizing target-like images, whose generator is consisted of an encoder and a decoder, our model only focus on extracting domain invariant representation, whose function is the same as encoder in the cycleGAN. Our feature generator can be easily applied to the cycleGAN by replacing the encoder of cycleGAN. We mentioned this question in 4.3 section, first paragraph, last line “There are other effective domain adaptation methods, such as Pixel-GAN (Bousmalis et al., 2017), DupGAN (Hu et al., 2018) and the NAGAN (Zhang et al., 2019), which are not used in the comparison because they focus on synthesizing target-like images according to source images to reduce the domain shift; our approach can be easily integrated into these works.”
> On the other hand, generating “translated” images with cycleGAN needs more computation, the training process is harder to control and the training time is longer than ours.
>  We’ll add more evidence to support our result (such as feature maps as suggested) in the final version. And cite  research on the OCT cross-domain adaptation for different OCT devices as suggested.
>
>
> - In Algorithm 1, what distribution follows the random weights in the initialization process?
>
> Reply: Thank you for your careful review, the weights were initialized by truncated normal distribution. We’ll add this to the final version.
>
>
> - Table 1 caption does not describe which metric is showcased. What is the meaning of the values in parentheses? This information is in the text but not in the caption of the Table.
>
> Reply: Thank you for you kind remind, we’ll revise this part in the final vision.
>
>
> Detailed Comments:
> “However, when the discriminator can perfectly distinguish the target from source representations, there is a gradient vanishing problems.” I think this is an oversimplification of GAN training difficulties. Additionally, the authors could also include a noisy loss signal, mode collapse, and the synchronization strength between the generator and discriminator.
>
> Reply: Thank you for your constructive comment, we’ll add more details about GAN training to the paper.
>
> - Please add details about the contrast enhancement used as data augmentation during training.
>
> Reply: Thank you for your remind, we’ll add details about data augmentation to the final version paper.
>
> - Figures 3. and 4. are very difficult to analyze correctly. Also, Tsne is a non-deterministic visualization technique that generates different visualizations and depends heavily on the perplexity hyperparameter. Given these limitations, I would say it is not advisable to use these visualizations to demonstrate that the method reduces the representations’ distance. I would recommend using feature maps from the generator (last conv layer before Fully connected layer) to showcase such reduction.
>
> Reply: Thank you for your good advice, we’ll add feature maps from the generator as suggeted to demonstrate the performance of our method.
>
> MINOR COMMENTS
> - Section "3. method" and "3.1. overview" are in lowercase
> - “…into a latent representationz = ”
> - “zwhich are defined not only at the source”
> - “to achieve the Nash Equilibrium faster (change the description).”
>
> Reply: We apologize for these errors. All minor comments raised by the reviewer will be corrected in the final version.

---

### Official Review · AnonReviewer3 · 2020-03-15
**A network for unsupervised domain adaptation with an application on OCT images**

**Rating:** 2
**Confidence:** 4

**Summary:**

This paper presents an adversarial domain adaptation method for retinopathy detection. The idea is to extract invariant and discriminative characteristics shared by different domains for the application of cross-domain OCT image classification task.  The application studied in this paper is interesting and potentially significant.


**Strengths:**

The application studied in this paper is interesting and potentially significant. The proposed network utilizes several components, which generally makes senses. Overall this paper is presented clearly.

**Weaknesses:**


1. The novelty of this paper is quite limited. There are many unsupervised domain adaptation methods, which could be directly applied to retinopathy detection. For example,
Conditional Adversarial Domain Adaptation
Adversarial Discriminative Domain Adaptation
Unsupervised Domain Adaptation with Adversarial Residual Transform Networks
DART: Domain-Adversarial Residual-Transfer Networks for Unsupervised Cross-Domain Image Classification
Unsupervised Domain Adaptation with Residual Transfer Networks

It is better to provide more discussion to these literatures.

2. Experiments seems to be quite insufficient and inconvincible. There is only one valid dataset for retinopathy detection. If this is the only available dataset, repeated experiments with cross-validation are needed.


**Detailed Comments:**

See above.

**Justification Of Rating:**

 The novelty of this paper is quite limited given a large number of adversarial domain adaptation methods.
And the experimental evaluation is insufficient, since there is only one dataset used and there is no significance test.

**Paper Type:**

validation/application paper

**Questions To Address In The Rebuttal:**

1. The novelty of this paper is quite limited. There are many unsupervised domain adaptation methods, which could be directly applied to retinopathy detection.  It is better to provide more discussion to these literatures.

2. Experiments seems to be quite insufficient and inconvincible. There is only one valid dataset for retinopathy detection. If this is the only available dataset, repeated experiments with cross-validation are needed.

**Special Issue:**

no

---

> ### Author Response · Authors · 2020-03-26
> **Reply to reviewer's comments**
>
> Thank you for your time and efforts in reviewing our paper. We clarify the issues as follows:
> 1. The novelty of this paper is quite limited. There are many unsupervised domain adaptation methods, which could be directly applied to retinopathy detection. It is better to provide more discussion to these literatures.
>
> Reply: We proposed a new method to reduce the discrepancy between OCT images from different devices by combining a Wasserstein distance estimator and a domain discriminator, as far as we know, this is the first work on domain adaptation research for retinopathy detection on OCT images. And we compared our method with the most relevant work WDGRL (Shen, Jian, et al. "Wasserstein distance guided representation learning for domain adaptation." Thirty-Second AAAI Conference on Artificial Intelligence. 2018.) and a new published work JDDA (Chen, Chao, et al. "Joint domain alignment and discriminative feature learning for unsupervised deep domain adaptation." Proceedings of the AAAI Conference on Artificial Intelligence. Vol. 33. 2019.).  Referring to your concern about the comparison between other domain adaptation methods and ours, we’ll add additional experiments to support our study as suggested in the final version.
>
> 2. Experiments seems to be quite insufficient and inconvincible. There is only one valid dataset for retinopathy detection. If this is the only available dataset, repeated experiments with cross-validation are needed.
>
> Reply: Under our experimental setup, it is impossible to perform cross validation, since labeled and unlabeled data are sampled from different distributions. Thus we evaluate all methods by empirically searching the parameter space for the optimal parameter settings, and report the best results of each method.

---

### Comment · Area_Chair1 · 2020-03-29
**Discussion phase starts**


Dear reviewers,

Thank you for your work and your reviews. We are now starting the discussion phase for MIDL. So far, looking at the current scores, this paper has received three weak accepts and one weak reject.

Could you check the authors responses to each of you, and participate to the discussions by posting messages either for confirming your previous position, notifying a change in your evaluation, or discussing some remaining unclear points. Thanks.

---

### Meta-Review · Area_Chair1 · 2020-04-07
**MetaReview of Paper334 by AreaChair1**

**Rating:** 3
**Recommendation For Accepted Papers:** Poster

**Metareview:**

Three out of four reviewers provide positive ratings, mainly based on appreciating the application value of DA for OCT data. I agree that this is a reasonable validation/application, and finding novel application for DA is of value to MIDL community.

The paper claims as "according to our knowledge, there is no research about domain adaptation on retinopathy detection from OCT images." In comments, R4 provided a reference "Reducing image variability across OCT devices with unsupervised unpaired learning for improved segmentation of retina” (Romo et al., 2019)", but ignored by the authors. I suggest the authors to carefully check the literature and avoid over-claim.





**Paper Type:**

validation/application paper

**Special Issue:**

no

---

### Decision · Program_Chairs · 2020-04-11

Accept